# Predicting Behavioral Intentions Related to Cervical Cancer Screening Using a Three-Level Model for the TPB and SCT in Nanjing, China

**DOI:** 10.3390/ijerph16193575

**Published:** 2019-09-24

**Authors:** Jianxin Zhang, Zimo Sha, Yuzhou Gu, Yanzhang Li, Qinlan Yang, Yuxuan Zhu, Yuan He

**Affiliations:** 1School of Humanities, Jiangnan University, Wuxi 214122, China; 2The Fourth School of Clinical Medicine, Nanjing Medical University, Nanjing 211166, China; 3School of Marxism, Nanjing Medical University, Nanjing 211166, China

**Keywords:** behavioral intentions related to cervical cancer screening, TPB, SCT, CCK, three-level model, path analysis

## Abstract

Objective: Exploring how the theory of planned behavior (TPB), social capital theory (SCT), cervical cancer knowledge (CCK), and demographic variables predict behavioral intentions (BI) related to cervical cancer screening among Chinese women. Methods: Self-administered questionnaires were distributed to 496 women, followed by a path analysis. Results: The three-level model was acceptable, χ^2^(26, 470) = 26.93, *p* > 0.05. Subjectively overcoming difficulties, support from significant others, screening necessity, and the objective promotion factor promoted BI, with effect sizes of 0.424, 0.354, 0.199, and 0.124. SCT and CCK promoted BI through TPB, with effect sizes of 0.262 and 0.208. Monthly income, education, age, and childbearing condition affected BI through TPB, SCT, and CCK, with effect sizes of 0.269, 0.105, 0.065, and −0.029. Conclusion: The three-level model systematically predicted behavioral intentions relating to cervical cancer screening.

## 1. Introduction

Cervical cancer is one of the two most common cancers with high mortality, whose incidence worldwide is the second highest for female malignant tumors, lower only than breast cancer [1,2]. Every year, more than 270,000 people die from it, and about 500,000 new cases are diagnosed globally. The highest incidence of carcinoma in situ ranges from 30 to 35 years old, and invasive carcinoma ranges from 45 to 55 years old. In recent years, its onset has shown a trend of younger age. Multiple sexual partners, pregnancies, and fertility are closely related to it [1,2]. Fortunately, through effective, simple, and economical screening, precancerous lesions can be identified early and treated to reduce the mortality rate very effectively. After the age of 25, it is necessary for women, especially those who have had sex or given birth, to have regular cervical cancer screening. Organized and well-designed screening services can significantly improve early diagnosis rate; the higher the screening coverage, the faster the mortality rate declines. Norway has only a 5% coverage of cervical cancer screening in the relevant population, resulting in only a 10% reduction in mortality in the last 30 years, while Finland and Iceland have 80% coverage and a 50% reduction in mortality. More than 80% of women in the United States were screened in 2010 [3].

The incidence of cervical cancer in China is increasing year by year, while the incidence in developed countries is decreasing. According to data from the Chinese National Cancer Registry, new cervical cancer cases accounted for 28.8% of the global total in 2015, six times higher than the rate in developed countries [4]. Screening should not have a scale effect, except when more than 70% of the target population is involved. The patients who should participate in screening tend not to participate, which greatly reduces screening effectiveness [5,6]. Since 2009, the Chinese State Council has carried out free cervical cancer screening for women aged 35–59 in a nationwide pilot, in order to improve screening coverage. However, mainly due to most women in China lacking willingness to participate in even free screening, only one fifth of women currently report having received cervical cancer screening [5,6], and data from Chinese Prevention and Control of Chronic Non-Communicable Diseases show that the coverage in rural pilot counties was only 20.94% [7].

Therefore, it is urgent to solve the problem of low screening rates. Previous studies have found that internal cognitive behavior factors, external social influencing factors, cervical cancer knowledge, and some demographic variables are the important influencing factors related to the behavior intentions for cervical cancer screening [8,9,10,11,12,13,14,15,16,17,18,19,20,21,22,23,24]. This information may provide direction and methods for interventions to investigate how these factors influence the behavior intentions related to cervical cancer screening among Chinese women.

### 1.1. Theory of Planned Behavior (TPB): Internal Cognitive Behavior Factors

The TPB focuses on internal cognitive behavior factors. As shown in Figure 1, the TPB can predict volitional behavioral intentions in terms of individuals’ perceptions of behavior. According to TPB, an individual’s actual behavior is influenced primarily by favorable or unfavorable perceptions of performing the behavior (behavioral attitude), perceptions of whether individuals important to him/her think the behavior should be performed (subjective norms), perception of the ease or difficulty of performing the behavior (perceived behavioral control), and plans to perform the behavior (behavioral intentions, BI).

Roncancio et al. (2015) revealed that the TPB was consistent with cervical cancer screening behavior [8]. Rutter (2000) validated that TPB effectively predicted women’s willingness and behavior related to breast cancer screening [9]. Henriksen, Guassora, and Brodersen (2015) revealed that acquaintances’ attitudes affected participation in breast cancer screening decision-making in Danish women [10]. Ogilvie et al. (2016) revealed that willingness to participate in cervical cancer screening was influenced by the calendar, positive attitude, and perceived behavioral control in Canadian women [11]. Guo, Zhang, and Wu (2011) revealed that subjective norms, perceived behavioral control, knowledge, behavioral attitude, history of benign breast diseases, income, education, occupation, and marital status degressively affected the BI of breast cancer prevention among Chinese women [12].

### 1.2. Social Capital Theory (SCT): External Social Influencing Factors

SCT pays attention to external social factors. Social capital refers to an individual’s ability to obtain scarce resources through purposeful actions, following social rules and embedded in the social structure, including social participation, social trust, social support, interpersonal interaction, and other factors.

Previous studies have revealed that social capital affects the BI of cervical cancer screening. Kristiansen et al. (2012) and Ogunwal et al. (2016) revealed that support from surrounding populations, male partners, and family members increased women’s participation in cervical cancer screening [13,14]. Madhivanan et al. (2016) demonstrated that family factors and education affected the use of cervical cancer screening services for Latin Miami immigrants [15]. The support of female family members was an important factor in promoting screening. Fear of screening leading to hysterectomy, cancer fatalism, embarrassment due to male doctor examinations, and fear of stigma of sexual disorders and sexually transmitted diseases were major obstacles. Jensen et al. (2016) revealed a significant positive correlation between low social support and non-participation in breast cancer screening [16]. Leader and Michael (2013) discovered that social capital perception affects breast cancer screening participation among American women [17]. Moudatsou et al. (2014) revealed that social capital promoted knowledge about and compliance with cervical cancer screening among rural Greek women [18].

### 1.3. Cervical Cancer Knowledge (CCK) and Demographic Variables

Studies have confirmed that CCK and demographic factors affect screening intentions and behavior. Several studies have revealed that CCK promotes women’s participation in screening [14,19,20,21]. Han et al. (2011) found that knowledge level, educational background, habits, and other factors influenced willingness and behavior related to cervical cancer screening in Chinese women [7]. Guo, Zhang, and Wu (2011) revealed that knowledge, income, education, occupation, and marital status affected breast cancer prevention intentions [12]. Taylor et al. (2002) and Lockwood-Rayermann (2004) discovered that a higher income promoted screening [22,23]. Katz and Hofer (1994) and Lockwood-Rayermann (2004) revealed that women’s participation in cervical cancer screening increased with their educational level [23,24].

### 1.4. Existing Research Problems and the Current Study Hypotheses

There are some issues in the abovementioned studies, as follows: (1) The TPB does not explain which factors influence internal cognitive behavioral factors of the TPB. Individual cognitive behavior is inevitably embedded in various social relationships and is constrained by social capital factors, such as social participation, social trust, social support, and interpersonal interactions. SCT can provide social reasons for the TPB. (2) SCT does not explain how the external social factors of SCT affect the BI of cervical cancer screening. External social factors must first influence an individual’s cognitive behavior factors, including their behavior attitude, subjective norms and perceptual behavior control, and then can influence the individual’s BI, thus affecting behavior. (3) The TPB focuses on the internal cognitive behavioral factors of BI, while SCT focuses on the external social factors. It is incomplete to explain and predict the BI of cervical cancer screening using only one theory, ignoring the relationship between the two theories [25]. Therefore, we must combine the two to build a system of internal and external causes for the BI of cervical cancer screening. (4) The mechanism by which CCK affects the BI of cervical cancer screening has not been explained by previous studies. Similar to social capital, CCK is a knowledge capital. Only through an individual’s cognitive behavioral system can it influence the BI. (5) The mechanism of demographic variables affecting the BI of cervical cancer screening has not been explained by previous studies. They can affect the BI through the other factors.

Therefore, the current study proposed a three-level hypothesis model: Demographic variables affect SCT and CCK (the third level), SCT and CCK affect the TPB (the second level), and the TPB affects the BI (the first level), see Figure 2. We compiled TPB, SCT, CCK, and demographic variable questionnaires to systematically explore how they influenced the BI of cervical cancer screening among Chinese women, so as to mobilize positive factors, overcome negative factors, and improve screening willingness and behavior in the future. The current study assumes that:

**Hypotheses (H1a)–(H1c).** 
*Behavioral attitude, subjective norms, and perceptual behavior control affect the BI of cervical cancer screening.*


**Hypotheses (H2a)–(H2c).** 
*Social capital affects the BI through influencing behavioral attitude, subjective norms and perceptual behavior control.*


**Hypotheses (H3a)–(H3c).** 
*CCK affects the BI through influencing behavioral attitude, subjective norms, and perceptual behavior control.*


**Hypotheses (H4a)–(H4c).** 
*Demographic variables affect behavioral attitude, subjective norms, and perceptual behavior control through influencing social capital, and finally affect the BI.*


**Hypotheses (H5a)–(H5c).** 
*Demographic variables affect behavioral attitude, subjective norms, and perceptual behavior control through influencing CCK, and finally affect the BI.*


## 2. Methods

### 2.1. Development of the Questionnaire

In the self-administered TPB questionnaire for cervical cancer screening, four dimensions were set up: Advantages and disadvantages of cervical cancer screening (4 items), support from surrounding individuals (8 items), overcoming difficulties (7 items), and screening intentions (3 items). A five-point (1–5) Likert scale was used to score “extreme disapproval, disapproval, general, approval, and extreme approval”. A “don’t know how to answer (N/A)” option was added, with a score of 0, indicating that the subjects lacked sufficient awareness and attention of the situation and psychology described by the item in real life.

In the self-administered SCT questionnaire, six dimensions were set up: Social participation (5 items), community trust (2 items), social support (3 items), social relations (3 items), interpersonal interaction (5 items), and interpersonal tolerance (2 items). The scoring method was the same as for the TPB questionnaire.

In the self-administered CCK questionnaire, four multiple-choice items were set up regarding screening methods, treatment methods, etiologies, and symptoms, and in each the subjects selected one option to get one point. Two single-choice items were set up on “early detection can cure” and “screening is free”, and 0–2 points were used for “disagreement (wrong), unclear (lack of understanding), or consent (correct)”. Because of the different scoring methods of each item, the data were converted to *T* scores and then analyzed.

In the demographic variables questionnaire, demographic variables such as age, education, monthly income, and childbearing condition, among others, were measured.

### 2.2. Subjects and Data Collection

The sample was chosen with completeness and validity considered. (1) As reported earlier, the highest incidences of cervical cancer were from 30 to 35 years old and from 45 to 55 years old. In recent years, its onset has shown a trend of younger age. Multiple sexual partners, pregnancies, and fertility are closely related to it. The age and fertility had to be seriously considered to ensure completeness in the sample chosen. Chinese marriage law stipulates that women must be at least 20 years old before they can get married and give birth. Therefore, in the current study, adult women between 20–60 years old in Nanjing City, Jiangsu Province in China, were chosen by convenient sampling to fill in self-administered questionnaires between June and October 2018. (2) Women with mental retardation, hysterectomies, or cervical cancer were excluded to ensure validity. Because women with mental retardation could not complete the questionnaires, women with hysterectomies did not need cervical cancer screening any more, and there was little reason for women with cervical cancer to have cervical cancer screening for early diagnosis and prevention, so the sample was representative of the total population. In total, 552 questionnaires were completed, but 46 were discarded as incomplete, resulting in a response rate of 91.67% (506/552). After 10 individuals’ data were deleted (choosing the same option), 496 valid subjects were left, whose main demographic variables are shown in Table 1.

### 2.3. Data Analysis

All analyses were conducted using SPSS17.0 and Amos 22.0 (IBM). The first step was to test the reliability and validity of the self-administered questionnaires. (1) The data of 496 subjects were randomly divided into two halves. (2) Exploratory factor analysis was undertaken to establish a theoretical model for one half of the data. The spindle factor decomposition method was used with iterations of no more than 100, and the oblique rotation method with Kaiser standardization was adopted. The internal consistency reliability was acceptable if the coefficient was no less than 0.6, and the discrimination was acceptable if the item-total correlation was no less than 0.3. (3) Confirmatory factor analysis was undertaken for the other half of the data to verify whether the theoretical model was in line with the actual data. (4) For all the data, the internal consistency reliability was acceptable if the coefficient was no less than 0.7, and the discrimination was acceptable if the item-total correlation was no less than 0.3. The second step was to create descriptive statistics for each dimension of the questionnaires and compare them with the absolute standard (that is, 3 points, general) to investigate if they were greater than 3 points. The third step was to add important demographic variables (age, education, monthly income, and childbearing condition) to test the three-level hypothesis model by path analysis (He, 2015; Grapentine and Teny, 2000; Guo, Zhang, and Wu, 2011) [12,26,27]. The statistical abbreviations used in the current study are shown in Table 2 [26,27].

### 2.4. Ethics Approval

Ethical approval to conduct this study was obtained from the Ethics Committee of Sir RunRun Hospital, Nanjing Medical University (grant number: 2019-SR-017). Oral informed consent from all the women who participated in the survey was obtained.

## 3. Results

### 3.1. TPB Questionnaire for Cervical Cancer Screening

The results of reliability and validity tests showed that the self-administered TPB questionnaire for cervical cancer screening had excellent reliability and validity: (1) For half of the data of the TPB questionnaire (*n* = 248), KMO = 0.85 > 0.5 and the sphericity test found *p* < 0.001, therefore the data were suitable for exploratory factor analysis. There were five factors whose initial eigenvalues were greater than 1. Screening necessity (behavioral attitude) included items 1, 3, and 4 with factor subordination values of 0.62, 0.87, and 0.72. Support from significant others (subjective norms) including items 5, 6, 7, 8, 9, and 10 with factor subordination values of 0.64, 0.71, 0.59, 0.82, 0.78, and 0.57. The objective promotion factor included items 2, 11, 12, 13, and 19 with factor subordination values of 0.65, 0.68, 0.73, 0.63, and 0.42. Subjectively overcoming difficulties (perceived behavioral control) included items 14, 15, 16, 17, and 18 with factor subordination values of 0.71, 0.79, 0.81, 0.84, and 0.86. Screening BI included items 20, 21, and 22 with factor subordination values of 0.81, 0.86, and 0.73. The internal consistency reliability of each dimension was no less than 0.6, and the discrimination of each item was no less than 0.3. (2) For the other half of the data (*n* = 248), confirmatory factor analysis was performed. After model modification, the factor loads of most items were above 0.55; the factor loads of items 1 and 19 were between 0.32 and 0.55. The model was acceptable, χ^2^(197, 248) = 450.54, *p* < 0.001, chi-square value of minimum sample/degrees of freedom (CMIN/DF) = 2.29 < 3. (3) The internal consistency reliability of the TPB (*n* = 496) was 0.88, except for item 19 (the number participating in cervical cancer screening in the past five years, which was very important); the discrimination of each item was greater than 0.3.

A single sample *t*-test showed that all dimensions were significantly greater than 3 points, indicating a good level, as shown in Table 3. Repeated variance analysis of the TPB dimensions was conducted. A sphericity test showed that the variance was not homogeneous, *p* < 0.05, and we then completed a Greenhouse correction and found that the TPB dimensions’ main effect was significant, *F*(3.63, 492) = 92.55, *p* < 0.001, η2p = 0.158. The paired comparison of the Bonferroni correction showed that screening necessity was greater than the other dimensions (*p*s < 0.01), support from significant others was greater than the other dimensions (*p*s < 0.001) except screening necessity, the objective promotion factor was smaller than the other dimensions (*p*s < 0.001), and there was no difference between subjectively overcoming difficulties and BI (*p* > 0.05).

### 3.2. SCT Questionnaire and CCK Questionnaire

The results of the reliability and validity tests showed that the self-administered SCT questionnaire had excellent reliability and validity: (1) For half of the data of the SCT questionnaire (*n* = 248), KMO = 0.89 > 0.5, and the sphericity test found *p* < 0.001; therefore, the data were suitable for exploratory factor analysis. There were five factors whose initial eigenvalues were greater than 1. Social participation included items 1, 2, and 3 with factor subordination values of 0.86, 0.96, and 0.79. Community trust included items 4, 5, 6, and 7 with factor subordination values of 0.57, 0.74, 0.72, and 0.73. Social support included items 8, 9, and 10 with factor subordination values of 0.70, 0.85, 0.57. Interpersonal interaction included items 11, 12, 13, 14, 15, 16, 17, and 18 with factor subordination values of 0.65, 0.65, 0.35, 0.86, 0.91, 0.78, 0.75, and 0.67. Interpersonal tolerance included items 19 and 20 with factor subordination values of 0.88 and 0.85. The internal consistency reliability of each dimension was no less than 0.6, and the discrimination of each item was no less than 0.3. (2) For the other half of the data (*n* = 248), confirmatory factor analysis was performed. The factor loads of most items were above 0.55; the factor load of item 13 was between 0.32 and 0.55. The model was acceptable, *χ*^2^(160, 248) = 390.86, *p* < 0.001, CMIN/DF = 2.44 < 3. (3) The internal consistency reliability of SCT (*n* = 496) was 0.89; the discrimination of each item was greater than 0.3.

The second-order exploratory factor analysis of the SCT questionnaire showed a common factor, indicating that the total average SCT score could be obtained [28]. The single-sample *t*-test showed that social participation was significantly less than 3 points; the other dimensions and the SCT score were significantly greater than 3 points. The scores are shown in Table 4.

In the CCK questionnaire, each item was independent to examine specific knowledge, so there was no need to undertake reliability and validity tests. Because of the different scoring methods, it was first converted to a *Z* score, then to a *T* score with *T* = 50 + 10 × *Z*. Finally, the average *T* score of the six items was regarded as the CCK score, *M* ± *SD* = 50.15 ± 6.90.

### 3.3. Demographic Variables Questionnaire

In the demographic variables questionnaire, for age, *M* ± *SD* = 1.88 ± 0.8; for education, *M* ± *SD* = 4.28 ± 1.03; for monthly income, *M* ± *SD* = 2.64 ± 1.06; and for childbearing condition, *M* ± *SD* = 0.98 ± 0.79. The demographic variable scores were grade data. Their significance is detailed in Table 1.

### 3.4. Path Analysis for the BI of Cervical Cancer Screening

Path analysis was performed to examine how the TPB, SCT, CCK, and demographic variables predicted the BI of cervical cancer screening. The parameter estimation is shown in Figure 3.

The fit parameters were as follows: χ^2^(26, 470) = 26.93, *p* > 0.05, CMIN/DF = 1.04 < 3; there was no significant difference between the data and the theoretical model. Absolute adaptation indexes: Root mean square error of approximation (RMSEA) = 0.01 < 0.08, root mean square residual (RMR) = 0.07 < 0.1, minimum of *F* (FMIN) = 0.05 < 2, goodness of fit index (GFI) = 0.99, adjusted goodness of fit index (AGFI) = 0.98; value-added adaptation indices: normative fit index (NFI) = 0.98, relative fit index (RFI) = 0.97, increase value fit index (IFI) = 1, Tucker–Lewis index (TLI) = 1, compare fit index (CFI) = 1. The information indices Akaike information index (AIC), consistent Akaike information index (CAIC), and expected cross-validation index (ECVI) of the theoretical model were the smallest. All indices represented good fit, and the model was acceptable.

Figure 3 and Table 5 show that the results model basically conformed to the three-level hypothetical model: (1) In the TPB, support from significant others, subjectively overcoming difficulties, screening necessity, and the objective promotion factor degressively, directly or indirectly, promoted the BI. Support from significant others had direct and indirect effects, subjectively overcoming difficulties and screening necessity only had a direct effect, and the objective promotion factor only had an indirect effect. They also interacted with each other. Support from significant others and the objective promotion factor degressively promoted subjectively overcoming difficulties, while support from significant others degressively promoted subjectively overcoming difficulties, screening necessity, and the objective promotion factor. (2) Both SCT and CCK promoted subjectively overcoming difficulties, support from significant others, screening necessity, and the objective promotion factor, thus indirectly or directly promoting the BI, in which social capital played a greater role and promoted CCK, and CCK also directly promoted the BI. (3) Education and monthly income promoted subjectively overcoming difficulties, support from significant others, screening necessity, and the objective promotion factor through promoting SCT and CCK, thus indirectly or directly promoting the BI. Among them, monthly income could promote the BI more and directly. However, childbearing condition indirectly hindered the BI through hindering CCK. Age indirectly promoted the BI through hindering CCK but promoting SCT. Childbearing condition and age had a small influence.

In concise terms, the three-level hypothesis model was basically established: (1) TPB promoted BI, which was the first level. (2) SCT and CCK promoted TPB, which was the second level. (3) Demographic variables affected SCT and CCK, which was the third level. Taking them together, important support from others, subjectively overcoming difficulties, monthly income, SCT, CCK, screening necessity, objective promotion factor, education, and age degressively promoted BI, but childbearing condition hindered BI.

## 4. Discussion

### 4.1. The First Level Affecting the BI: The TPB Internal Cognitive Behavior Factors

Path analysis showed that support from significant others (subjective norms), subjectively overcoming difficulties (perceived behavioral control), screening necessity (behavioral attitude) and the objective promotion factor degressively promoted BI, which indicated that the TPB was suitable for predicting the BI of cervical cancer screening [8,9,10,11,12]. It proved that hypotheses H1a–H3c were consistent with real data.

Subjective norms, perceived behavioral control, and behavioral attitude played greater roles, while the objective promotion factor played a smaller role. The current study also revealed that they interacted with each other. Support from significant others and the objective promotion factor degressively promoted subjectively overcoming difficulties, while support from significant others degressively promoted subjectively overcoming difficulties, screening necessity, and the objective promotion factor, which indicated that subjectively overcoming difficulties was an important mediating variable for other variables promoting BI and that support from significant others was an important independent variable. The reason for this might be that for women, perceived support from significant others was one of their advantages (Zhang, Zhang, and Li, 2015) [29]. The objective promotion factor played a relatively small role, probably because it was not developed at this stage. The evidence was that the objective promotion factor was smallest among the five TPB dimensions (see Figure 2 and Table 5).

Therefore, it is important for intervenors to make sure women perceive support from significant others and to improve their confidence of overcoming difficulties, the latter of which will promote the effects of other variables. Since the objective promotion factor was lowest, it also needs to be further developed. For women, they need to develop these factors and their perception of these factors, so as to increase BI.

### 4.2. The Second Level Affecting the BI: SCT External Social Influencing Factors and CCK

SCT and CCK influenced the BI of cervical cancer screening, which was basically consistent with the existing research [13,14,15,16,17,18,19,20,21]. However, path analysis revealed that both SCT and CCK first promoted support from significant others, subjectively overcoming difficulties, screening necessity, and the objective promotion factor of the TPB, thus indirectly or directly promoting the BI, which proved H2a–H3c hypotheses. The current study, for the first time, combined the TPB and SCT. It confirmed the external SCT really needed to influence the BI through promoting the internal TPB to construct a system of internal and external factors, which was more comprehensive, more revealing (revealing the influence course), and therefore more predictive than the existing TPB or SCT studies. SCT played a greater role in promoting the BI than CCK, and it also indirectly promoted BI by promoting CCK. CCK could both directly promote the BI and indirectly promote the BI through promoting the TPB, indicating that the impact mechanism was complex.

Intervenors can directly improve women’s CCK, but it is hard to directly improve women’s SCT, which can be indirectly improved with help at the policy, economic, and educational levels. Intervenors can make women realize the importance of SCT in order to improve it, which can also indirectly increase CCK. For women, they also need to realize the importance of SCT to improve it, so as to increase BI.

### 4.3. The Third Level Affecting the BI: Monthly Income, Education, Age, and Childbearing Condition

Monthly income, education, age, and childbearing condition affected the BI of cervical cancer screening, which was basically consistent with previous studies [6,12,22,23,24]; however, it also revealed the mechanism by which they affected SCT and CCK, thereby affecting the TPB, and then degressively affecting the BI. This proved hypotheses H4a–H5c and proved that the three-level hypothesis model was valid.

Monthly income and education indirectly promoted support from significant others, subjectively overcoming difficulties, screening necessity, and the objective promotion factor by promoting SCT and CCK, thus indirectly or directly promoting the BI. Monthly income also directly promoted the BI and support from significant others. Education promoted monthly income, but monthly income promoted BI more and more directly than education. There were three possible reasons for this: (1) The higher the monthly income, the stronger the woman’s ability, and the greater the woman’s social capital. The evidence was that monthly income promoted SCT, and the regression weight was 0.204 (see Figure 2 and Table 5). (2) The higher the monthly income, the more important position in the family the woman might occupy (Sun, Zheng, and Xu, 2018) [30], the more resources (money, SCT, and CCK) for cervical cancer screening they could get (Zhang and Wu, 2011) [12], and the more attention they might pay to their health. (3) Monthly income might directly determine affordability of screening, which is not all free for all women in China. The causes need to be determined, which will lead to new research.

Age and childbearing condition had less influence on the BI than monthly income and education. Age indirectly promoted the BI by hindering CCK but promoting social capital. Childbearing condition hindered the BI indirectly by hindering CCK. This was inconsistent with common sense (Zhu and Zhang, 2014) [31]. There were three possibilities for this finding: (1) Childbearing condition was promoted by age; the regression weight was 0.529 (see Figure 2 and Table 5). The older the woman’s age, the lower the average educational level was, and the regression weight was −0.372 (see Figure 2 and Table 5), so cervical cancer screening might be not known about or paid attention to. This was assumed to be due to the special educational history in China. (2) The more births a woman has undergone, the more they might think their uterus was healthy, or the less they and their family might think it was necessary to undertake cervical cancer screening because they had enough children. (3) The more births a woman has undergone, the more energy they might focus on children while neglecting themselves. The causes need to be determined, which will lead to new research. The childbearing condition also directly promoted the objective promotion factor, but directly hindered screening necessity.

Intervenors need to pay more attention to and increase advocacy and support for women with multiple births, who are at greater risk of having cervical cancer [1,2], but who are less willing to participate in screening. Women with multiple births especially need to realize the cervical cancer risk of multiple births and the necessity and effectiveness of cervical cancer screening. Of course, monthly income and education for women need to rise, which requires help at the policy, economic, and educational levels.

### 4.4. The Three-Level Model: The Sequential and Comprehensive Influence of Various Factors

In conclusion, demographic variables affected SCT and CCK, SCT and CCK promoted the TPB, and the TPB promoted the BI. The three-level hypothesis model was basically established. Support from significant others, subjectively overcoming difficulties, monthly income, SCT, CCK, screening necessity, the objective promotion factor, education, and age degressively promoted the BI, but childbearing condition hindered the BI. Therefore, research and policies should not only use a certain theory such as the TPB or SCT but should comprehensively examine the internal relationships and levels between various theories and factors [25], so as to systematically predict and increase women’s screening enthusiasm.

The three-level prediction model can be used as a reference for future intervention research: (1) The TPB factors were the direct positive and decisive factors for the BI, with relatively high promotion levels [12]. Almost all other factors should influence the BI through the TPB. Subjectively overcoming difficulties was an important mediating variable, and support from significant others was an important independent variable. Therefore, the interventions need to be carried out around TPB. It will be very effective for increasing BI and influence of other factors by developing women’s perception of subjective norm, perceived behavioral control, behavioral attitude, and objective promotion factor. (2) The SCT and CCK indirectly promoted the BI through promoting the TPB, with relatively moderate promotion values. They were also important mediating variables for demographic variables. Therefore, interventions need to enhance SCT and CCK, which should also promote the TPB and magnify the impact of demographic variables. (3) Monthly income, education, age, and childbearing condition affected SCT and CCK, thereby affecting the TPB, and then degressively affected the BI, with a relatively low degree of influence. Interventions need to enhance monthly income and education and increase support for women with more children. Because demographic variables are difficult to change in the short term, long-term and multifaceted support is required. (4) It may be more effective to intervene by synthesizing the above three level factors simultaneously or step by step than changing a single factor. This requires the cooperation of women and their social relations, more comprehensive and feasible intervention programs, more interventionists, more money and time, and more comprehensive laws and policies. For women, they can also synthesize the above three levels factors to increase BI. Of course, the methods and effects of synthesized interventions need to be explored by new intervention researches.

## 5. Conclusions

The three-level model was established, in which demographic variables affect SCT and CCK, SCT and CCK affect the TPB, and the TPB affects behavioral intentions related to cervical cancer screening. Support from significant others, subjectively overcoming difficulties, monthly income, SCT, CCK, screening necessity, the objective promotion factor, education, and age degressively promoted the BI, but the childbearing condition hindered the BI. Future interventions should synthesize the three level factors simultaneously, or step by step.

## Figures and Tables

**Figure 1 ijerph-16-03575-f001:**
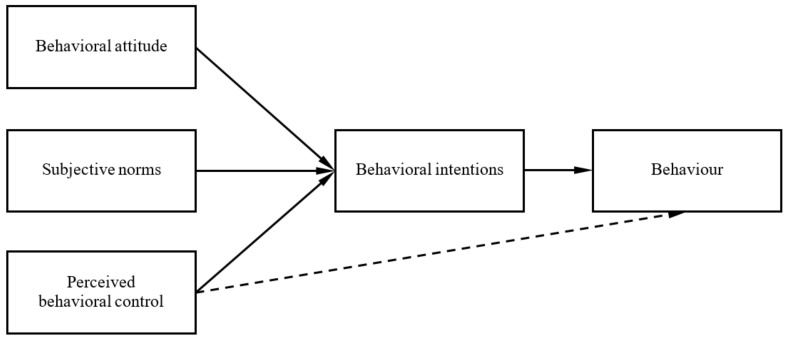
The theory of planned behavior (TPB) model of cancer screening.

**Figure 2 ijerph-16-03575-f002:**
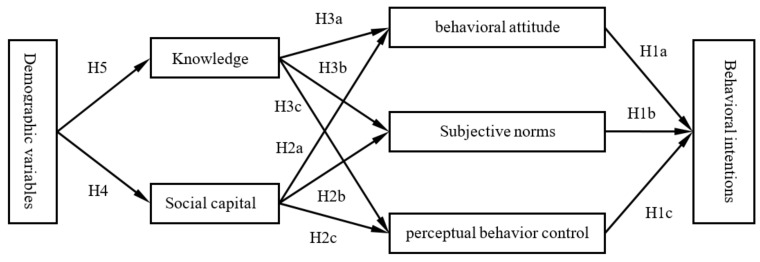
The three-level hypothesis model of the behavioral intentions (BI) of cervical cancer screening.

**Figure 3 ijerph-16-03575-f003:**
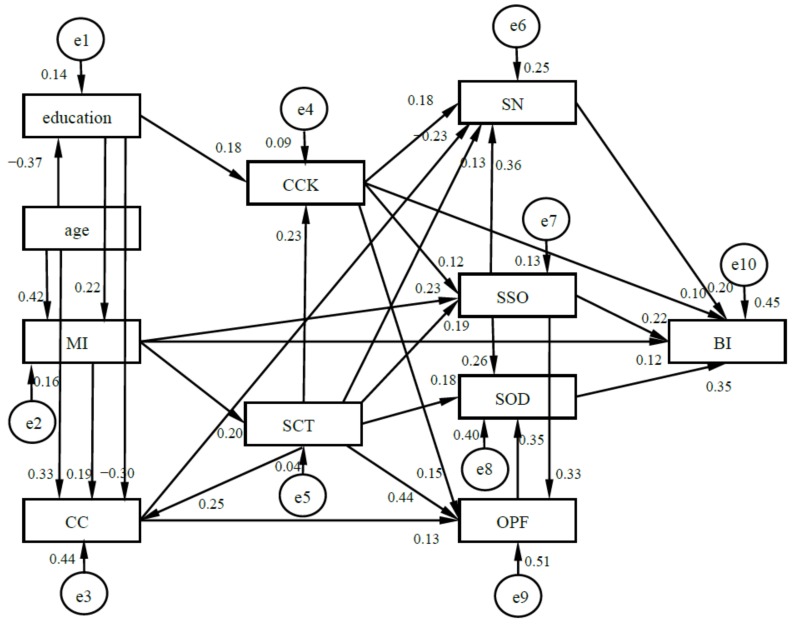
Results model of the BI of cervical cancer screening (*n* = 496). Note: MI, CC, SN, SSO, SOD, and OPF are the abbreviations of monthly income, childbearing condition, screening necessity, support from significant others, subjectively overcoming difficulties, and the objective promotion factor.

**Table 1 ijerph-16-03575-t001:** Demographic variables (*n* = 496).

Variables	Values					
Age	20–30	31–40	41–50	51–60		
Number	178	210	96	12		
Proportion	0.36	0.42	0.19	0.02		
Education	primary school and below	junior school	high school/technical school	university or college	undergraduate	master’s degree or above
Number	2	40	60	131	243	20
Proportion	0	0.08	0.12	0.26	0.49	0.04
Monthly income	1000 yuan and below	1001–3000 yuan	3001–5000 yuan	5001–8000 yuan	8001 yuan and above	
Number	91	108	201	79	17	
Proportion	0.18	0.22	0.41	0.16	0.03	
Childbearing condition	0	1	2	3	above 4	
Number	150	216	121	9	0	
Proportion	0.3	0.44	0.24	0.02	0	

Note: Each variable was graded according to the assignment of 1, 2, 3.

**Table 2 ijerph-16-03575-t002:** Statistical abbreviations.

Statistical Abbreviation	The Full Name	Meaning	Function	Standard
KMO	Kaiser-Meyer-Olkin	ratio of simple correlation coefficients and partial correlation coefficients among all variables	measure whether the variables are suitable for exploratory factor analysis	>0.5
CMIN/DF	chi-square value of minimum sample/degrees of freedom	absolute adaptation indexes	measure the difference degree between the data and the theoretical model	<3
RMSEA	root mean square error of approximation	absolute adaptation indexes	same as above	<0.08
RMR	root mean square residual	absolute adaptation indexes	same as above	<0.1
FMIN	minimum of *F*	absolute adaptation indexes	same as above	<2
GFI, AGFI	goodness of fit index, adjusted goodness of fit index	absolute adaptation indexes	proportion that theoretical model can explain the data	>0.9
NFI, RFI, IFI, TLI, CFI	normative fit index, relative fit index, increase value fit index, Tucker–Lewis index, compare fit index	value-added adaptation indices	measure the ratio between the data and the theoretical model	>0.9
AIC, CAIC, ECVI	Akaike information index, consistent Akaike information index, expected cross-validation index	information indices	measure how simply and effectively the model interprets the information of the data	The theoretical model value is smaller than both the saturation model value and independent model value

**Table 3 ijerph-16-03575-t003:** TPB dimension scores (*n* = 496).

Dimension	*M* ± *SD*	*t*	Cohen’s *d*
screening necessity	4.02 ± 0.7	32.38 ***	1.46
support from significant others	3.88 ± 0.84	23.16 ***	1.05
objective promotion factor	3.29 ± 0.95	6.85 ***	0.31
subjectively overcoming difficulties	3.66 ± 0.88	16.76 ***	0.75
BI	3.73 ± 0.81	20.25 ***	0.9

Note: *** *p* < 0.001. The *t* value was the result of a single sample *t*-test, with 3 points as the comparison value.

**Table 4 ijerph-16-03575-t004:** Social capital theory (SCT) dimension scores (*n* = 496).

Dimension	*M* ± *SD*	*t*	Cohen’s *d*	Common Factor Dependency
social participation	2.43 ± 1.11	–11.37 ***	−0.51	0.69
community trust	3.44 ± 0.82	11.96 ***	0.54	0.74
social support	3.63 ± 0.74	19.02 ***	0.85	0.65
interpersonal interaction	3.08 ± 0.8	2.10 *	0.1	0.87
interpersonal tolerance	3.56 ± 0.84	14.91 ***	0.67	0.58
SCT	3.23 ± 0.66	7.63 ***	0.35	-

Note: * *p* < 0.05, *** *p* < 0.001. The t value was the result of a single sample t-test, with 3 points as the comparison value.

**Table 5 ijerph-16-03575-t005:** Path coefficients decomposition of the BI of cervical cancer screening (*n* = 496).

Causal Variables	Outcome Variables	Total Effect	Direct Effect	Indirect Effect
age	education	−0.372	−0.372	0
	monthly income	0.345	0.425	−0.08
	SCT	0.07	0	0.07
	cervical cancer knowledge (CCK)	−0.052	0	−0.052
	support from significant others	0.088	0	0.088
	childbearing condition	0.529	0.335	0.194
	objective promotion factor	0.12	0	0.12
	subjectively overcoming difficulties	0.077	0	0.077
	screening necessity	−0.09	0	−0.09
	BI	0.065	0	0.065
education	monthly income	0.215	0.215	0
	SCT	0.044	0	0.044
	CCK	0.193	0.183	0.01
	support from significant others	0.081	0	0.081
	childbearing condition	−0.252	−0.302	0.051
	objective promotion factor	0.043	0	0.043
	subjectively overcoming difficulties	0.043	0	0.043
	screening necessity	0.127	0	0.127
	BI	0.105	0	0.105
monthly income	SCT	0.204	0.204	0
	CCK	0.046	0	0.046
	support from significant others	0.278	0.234	0.044
	childbearing condition	0.236	0.185	0.05
	objective promotion factor	0.218	0	0.218
	subjectively overcoming difficulties	0.183	0	0.183
	screening necessity	0.08	0	0.08
	BI	0.269	0.122	0.147
SCT	CCK	0.226	0.226	0
	support from significant others	0.214	0.188	0.026
	childbearing condition	0.247	0.247	0
	objective promotion factor	0.578	0.442	0.136
	subjectively overcoming difficulties	0.432	0.175	0.257
	screening necessity	0.192	0.131	0.061
	BI	0.262	0	0.262
CCK	support from significant others	0.117	0.117	0
	objective promotion factor	0.191	0.152	0.038
	subjectively overcoming difficulties	0.097	0	0.097
	screening necessity	0.223	0.181	0.042
	BI	0.208	0.104	0.104
support from significant others	objective promotion factor	0.327	0.327	0
	subjectively overcoming difficulties	0.37	0.256	0.114
	screening necessity	0.356	0.356	0
	BI	0.424	0.222	0.202
childbearing condition	objective promotion factor	0.129	0.129	0
	subjectively overcoming difficulties	0.045	0	0.045
	screening necessity	−0.228	−0.228	0
	BI	−0.029	0	−0.029
objective promotion factor	subjectively overcoming difficulties	0.349	0.349	0
	BI	0.124	0	0.124
subjectively overcoming difficulties	BI	0.354	0.354	0
screening necessity	BI	0.199	0.199	0

Note: *p* < 0.01 for all regression weights of each path.

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
