# Peer review of "Predicting Behavioral Intentions Related to Cervical Cancer Screening Using a Three-Level Model for the TPB and SCT in Nanjing, China"

_ijerph, 2019, doi:10.3390/ijerph16193575_

Round 1
Reviewer 1 Report
very good study design and presentation
Author Response
Response to Reviewer 1 Comments
Point 1: very good study design and presentation
Response 1: We appreciate your comments, which will encourage us to do better job in the future research. Thank you very much for your useful suggestions and comments!
Reviewer 2 Report
In my opinion that article is interesting but is not easy to perceive. I suggest some changes: first of put a list of abbreviations (many of them have not been explained e.g.KMO, RMSEA, CMIN/DF....). Also the authors write: "previous studies found (..) line 47, but not provide references in this place.
The second paragraph include Methods : 2.1. Subjects- but it is not subject but methodology and materials. In this place the authors should explain thecriteria of inclusion and exclusion. Was the group the representative? How the authors choose the sample?
The paragraph 2.2. is not materials but this is still a methods.
The title Discussion should be changed because it is rather a discussion of the results.
Conclusions- more practical conclusions rather than synthetic repetition of results.
Author Response
Response to Reviewer 2 Comments
Thank you very much for your useful suggestions and comments, which guided us to improve the quality of this article. We have revised this article following your opinions using the "Track Changes". We look forward to your further review.
Comments and Suggestions for Authors
Point 1: In my opinion that article is interesting but is not easy to perceive. I suggest some changes: first of put a list of abbreviations (many of them have not been explained e.g.KMO, RMSEA, CMIN/DF....).
Response 1:Thank you for your valuable comments! The explanation of statistical abbreviations used in the current study was added in Table 2(He, 2015; Grapentine & Teny, 2000). Please see below and the revised article. (Line 239).
Table 2. Statistical abbreviations.
|
Statistical abbreviation |
The full name |
Meaning |
Function |
standard |
|
KMO |
Kaiser-Meyer-Olkin |
ratio of simple correlation coefficients and partial correlation coefficients among all variables |
measure whether the variables are suitable for exploratory factor analysisare |
> 0.5 |
|
CMIN/DF |
chi-square value of minimum sample/Degrees of freedom |
absolute adaptation indexes |
measure the difference degree between the data and the theoretical model |
< 3
|
|
RMSEA |
root mean square error of approximation |
absolute adaptation indexes |
same as above |
< 0.08 |
|
RMR |
root mean square residual |
absolute adaptation indexes |
same as above |
< 0.1 |
|
FMIN |
minimum of F |
absolute adaptation indexes |
same as above |
<2 |
|
GFI, AGFI |
goodness of fit index, adjusted goodness of fit index |
absolute adaptation indexes |
proportion that theoretical models can explain the data |
> 0.9 |
|
NFI, RFI, IFI, TLI, CFI |
normative fit index, relative fit index, increase value fit index, Tucker-Lewis index, compare fit index |
value-added adaptation indices |
measure the ratio between the data and the theoretical model |
> 0.9 |
|
AIC, CAIC, ECVI |
akaike information index, consistent akaike information index, expected cross-validation index |
information indices |
measure how simple and effective the model interprets the information of the data |
The theoretical model value is smaller than both saturation model value and independent model value |
References:
He X Q. Multivariate Statistical Analysis [M]. 4th Edition. Beijing, Renmin University Press(2015). Grapentine, Teny. Path Analysis VS. Structural Equation Modeling. MarketingResearch, 10-20(2000).
Point 2: Also the authors write: "previous studies found (..) line 47, but not provide references in this place.
Response 2: This sentence was intended to summarize and introduce the influencing factors to be discussed. The content and references were shown as below 1.1-1.3. We have added these references in this paragraph. ([8-24],see references 8-24)(Line 58)
Point 3: The second paragraph include Methods: 2.1. Subjects- but it is not subject but methodology and materials. In this place the authors should explain the criteria of inclusion and exclusion. Was the group the representative? How the authors choose the sample? The paragraph 2.2. is not materials but this is still a methods.
Response 3: Thank you for reminding us! We are very sorry for neglecting these problems. The the criteria of inclusion and exclusion for sample was added. (Line 183-197). Because the sample only contained women in Nanjing city, the title of this article was changed to “Predicting behavioral intention of cervical cancer screening by three-level model on TPB and SCT in Nanjing, China”. (Title). The junior titles of Methods were changed. (Line 161 and Line 182) Please see below and the revised article.
2.1. Development of the questionnaires
2.2. Subjects and data collection
The sample was chosen with completeness and validity considered. (1)As reported earlier, the highest incidences of cervical cancer were from 30 to 35 years old and from 45 to 55 years old. In recent years, its onset has shown a trend of younger age. Multiple sexual partners, pregnancies, and fertility are closely related to it. The age and fertility had to be seriously considered to ensure completeness in the sample chosen. Chinese marriage law stipulates that women must be at least 20 years old before they can get married and give birth. Therefore in the current study, adult women between 20–60 years old in Nanjing city, Jiangsu Province in China were chosen by convenient sampling to fill in self-administered questionnaires between June and October 2018. (2)Women with mental retardation, hysterectomies, or cervical cancer were excluded to ensure validity. Because women with mental retardation could not complete the questionnaires; women with hysterectomies did not need cervical cancer screening any more, and there was little meaning for women with cervical cancer to have cervical cancer screening for early diagnosis and prevention. So the sample was representative of the total population. (Line 183-197).
Point 4: The title Discussion should be changed because it is rather a discussion of the results.
Response 4: The titles and content of Discussion were changed. (Line 374,400,419 and 463). More evidences and references were added to make deeper discussion. Please see the revised article. (Line 381-399, Line 414-418, Line 429-440, Line 457-462 , Line 464-468 and Line 476-501).
Point 5: Conclusions- more practical rather than synthetic repetition of results.
Response 5: The conclusions were changed to be more practical. Please see the revised article.(Line 503-510).
Reviewer 3 Report
the article explores how the theory of planned behaviour, social capital theory, cervical cancer knowledge and demographic variables can predict behavioural intention of cervical cancer screening.
The introduction is well written and very clear. The subject is of great interest, the approach is very original and the methodology is adequate. however, the impact of the article could be improved with a better presentation of results. presentation of results is a mess, messages are not clear and impacts for patients or decision-makers are not explained. Interpretation of the results in the discussion is far-fetched and needs to be rewritten. English needs to be revised too.
line 147: "Lickett" should be replaced by "Likert"
Author Response
Response to Reviewer 3 Comments
Thank you very much for your valuable comments, which guided us to improve the quality of this article. We have revised this article following your opinions using the "Track Changes", and look forward to your further review.
Point 1: The introduction is well written and very clear. The subject is of great interest, the approach is very original and the methodology is adequate. however, the impact of the article could be improved with a better presentation of results. Presentation of results is a mess, and messages are not clear.
Response 1: Thank you very much for your comprehensive and incisive advice! We did the following efforts to make the results more orderly and clear.
(1) In 2. Methods, the explanation of statistical abbreviations used in the current study was added in Table 2 to make the results easier to perceive.(Table 2, Line 239). The steps of data analysis were introduced in detail. (Line 223-238)
(2) In 3. Results, repeated variance analysis of TPB dimensions was made to get more information. (Line 267-274) .The analysis of CCK questionnaire was changed to be with SCT Questionnaire to make “3.2. SCT Questionnaire and CCK Questionnaire”. (Line 278). Because both of them were the second level in the following path analysis. The analysis of demographic variables questionnaire was separated out to make “3.3. Demographic Variables Questionnaire”.(Line 304). Because it was the third level in the following path analysis.
(3)According to the requirements of path analysis(He, 2015; Grapentine & Teny, 2000; Guo, Zhang, & Wu, 2011), the figure of result model of parameter estimation, fit parameters and decomposition report of path coefficients were presented.(Line 236-238). Figures and tables and their textual exposition were as brief as possible. In terms of concise and generalization, a brief summary of the path analysis results was added after Table 5 in 3. Results.(Line 364)
References:
Guo Sf, Zhang C P, Wu J. Path analysis of influencing factors of breast cancer prevention behavioral intention in community women. Nursing Research, 25(11), 2909- 2911(2011). He X Q. Multivariate Statistical Analysis [M]. 4th Edition. Beijing, Renmin University Press(2015). Grapentine, Teny. Path Analysis VS. Structural Equation Modeling. MarketingResearch, 10-20(2000).
Point 2: Impacts for patients or decision-makers are not explained. Interpretation of the results in the discussion is far-fetched and needs to be rewritten.
Response 2: Thank you for the comments. We are very sorry for neglecting these problems. The discussion was rewritten. The order was first discourses and then synthesis, which was the first level, the second level, the third level, and their interaction and integration. (Line 374-501). Impacts for patients or decision-makers were added in 4.3 and 4.4 of the section of Discussion.(Line 457-462, Line 476-479 and Line 499-501). To avoid far-fetched discussion, more evidences and references were added to support the discussion. (Line 373-501 and Line 579-589)If it was just speculation, we indicated that it need to be determined by new researches. Please see the revised article.
Point 3: English needs to be revised too.
Response 3: English was revised by an English editing service of MDPI.
Point 4: Line 147: “Lickett” should be replaced by “Likert”.
Response 4: “Lickett” was replaced by “Likert”.(Line 165).